# Examining the potential impacts of a coastal renourishment project on the presence and abundance of *Escherichia coli*

Jordan A. Lewis[1,2]*, Victoria J. Frost[1], Matthew J. Heard[3]

**1** Department of Biology, Winthrop University, Rock Hill, South Carolina, United States of America,
**2** Department of Ecology and Evolutionary Biology, Yale University, New Haven, Connecticut, United States of America, **3** Department of Biology, Belmont University, Nashville, Tennessee, United States of America

* jordan.lewis@yale.edu

**Data Availability Statement:** All data files associated with this manuscript are available from the Dryad database https://doi.org/10.5061/dryad. zgmsbcchp.

## Abstract

Erosion poses a significant threat to oceanic beaches worldwide. To combat this threat, management agencies often utilize renourishment, which supplements eroded beaches with offsite sand. This process can alter the physical characteristics of the beach and can influence the presence and abundance of microbial communities. In this study, we examined how an oceanic beach renourishment project may have impacted the presence and abundance of *Escherichia coli* (*E. coli*), a common bacteria species, and sand grain size, a sediment characteristic that can influence bacterial persistence. Using an observational field approach, we quantified the presence and abundance of *E. coli* in sand (from sub-tidal, intertidal, and dune zones on the beach) and water samples at study sites in both renourished and non-renourished sections of Folly Beach, South Carolina, USA in 2014 and 2015. In addition, we also measured how renourishment may have impacted sand grain size by quantifying the relative frequency of grain sizes (from sub-tidal, intertidal, and dune zones on the beach) at both renourished and non-renourished sites. Using this approach, we found that *E. coli* was present in sand samples in all zones of the beach and at each of our study sites in both years of sampling but never in water samples. Additionally, we found that in comparison to non-renourished sections, renourished sites had significantly higher abundances of *E. coli* and coarser sand grains in the intertidal zone, which is where renourished sand is typically placed. However, these differences were only present in 2014 and were not detected when we resampled the study sites in 2015. Collectively, our findings show that *E. coli* can be commonly found in this sandy beach microbial community. In addition, our results suggest that renourishment has the potential to alter both the physical structure of the beach and the microbial community but that these impacts may be short-lived.

## Introduction

Erosion at marine beaches is a major global issue, and it is estimated that ~70% of oceanic coastlines worldwide are experiencing erosion [1]. For instance, over half of all marine beaches in the United States are categorized as critically eroded [2]. To combat this threat, land

**Funding:** This research was supported in part by grant P20GM103499 (SC INBRE) from the National Institute of General Medical Sciences (https://www.nigms.nih.gov/), National Institutes of Health (https://www.nih.gov/) (MH), Winthrop University Ronald E. McNair Post-Baccalaureate Achievement Program (P217A130111) from the US Department of Education (https://www.ed.gov/) (JL), the Rob Fisher Endowment of Belmont University (MH), and the National Science Foundation Graduate Research Fellowship Program (2017246734) (https://www.nsfgrfp.org/) (JL). We would also like to thank the National Science Foundation Postdoctoral Research Fellowship in Biology (https://www.nsf.gov/) (JL) for support during the writing and editing of this manuscript. The funders of this project played no role in its design, the collection or analysis of related data, the preparation of this manuscript, or the decision to publish.

**Competing interests:** The authors have declared that no competing interests exist.

managers and municipalities often use oceanic beach renourishment, which involves dredging sand from an offsite location and using that sand to replace lost surface area on the beach [3].

While renourishment can be advantageous economically compared to other methods (e.g., building hardscape projects like sea walls and jetties) (4), it can also physically and biologically impact beaches. Physically, renourishment can change the size and mineralogy of sand [4], sand porosity [5], turbidity of bathing waters [4], and sand surface temperature [6], depending on the source of new material. The impact of these physical changes on oceanic beaches can also alter the presence and abundance of species that persist in this ecosystem. The majority of biological research characterizing the impacts of renourishment have focused on macro-organisms [7–14]. However, research suggests that renourishment can drive changes in survival among microbial species as well [15,16]. Given that beaches can harbor a wide variety of microorganisms, and concerns related to beaches serving as reservoirs or vectors of infection [17], understanding how renourishment alters these communities has potentially significant implications for public health and monitoring strategies, in addition to our understanding of the basic ecology of these environments [18,19].

In this study, we examined how a coastal renourishment project may have influenced the presence and abundance of a common bacterial species, *Escherichia coli*, at a recently renourished oceanic beach in South Carolina, USA. While *E. coli* has not historically been used for environmental microbiology studies in marine ecosystems due to its conceived inability to tolerate salt stress [20], *E. coli* can show substantial variation in their ability to persist physical stressors in secondary environments [21–23]. Environmentally naturalized populations of *E. coli* have been identified [24], and research has supported the idea that the sediment microenvironment may be advantageous for *E. coli* in several ways [22,25]. Indeed, more recent studies suggest that *E. coli* can be viable in marine beach sediment and may be a relatively common component of the microbial community of beach sand [26–34]. The existence of naturalized populations, the numerous studies that have isolated *E. coli* from marine sand, and the ability of *E. coli* to vary widely in their ability to survive outside of the vertebrate gastrointestinal system suggests that *E. coli* may be a more habitual component of these ecosystems than previously hypothesized.

To assess the potential impacts of renourishment on *E. coli*, we conducted an observational study at Folly Beach, SC, USA, that examined how the presence and abundance of *E. coli* varied between renourished and non-renourished sections of the beach in June 2014 and June 2015. In total, we collected more than 240 samples of sand and water in June 2014 and June 2015 from 30 study sites (20 renourished sites and 10 non-renourished sites) along the beach (Fig 1). At each location, we sampled three zones of the beach based on tide level (subtidal, intertidal, dunes) and enumerated *E. coli* samples from each zone and from the water. The dunes are sometimes referred to as the "supratidal" zone in other studies or the areas of the beach above the spring high tide [35–37]. Lastly, we investigated how renourishment may have impacted sand grain sizes in each of the three beach zones in renourished and non-renourished sections to assess one possible mechanism for any observed differences in *E. coli* presence and abundance.

## Materials and methods

### Description of field sites

We conducted this study at Folly Beach, South Carolina, USA. Folly Beach is a 10 km long and 0.5 km wide barrier island approximately ~19 km south of Charleston, South Carolina, USA, and is estimated to account for around 3% of the total beach tourism in the state [39]. The island is a barrier island with characteristics typical of those found in the coastal marine areas

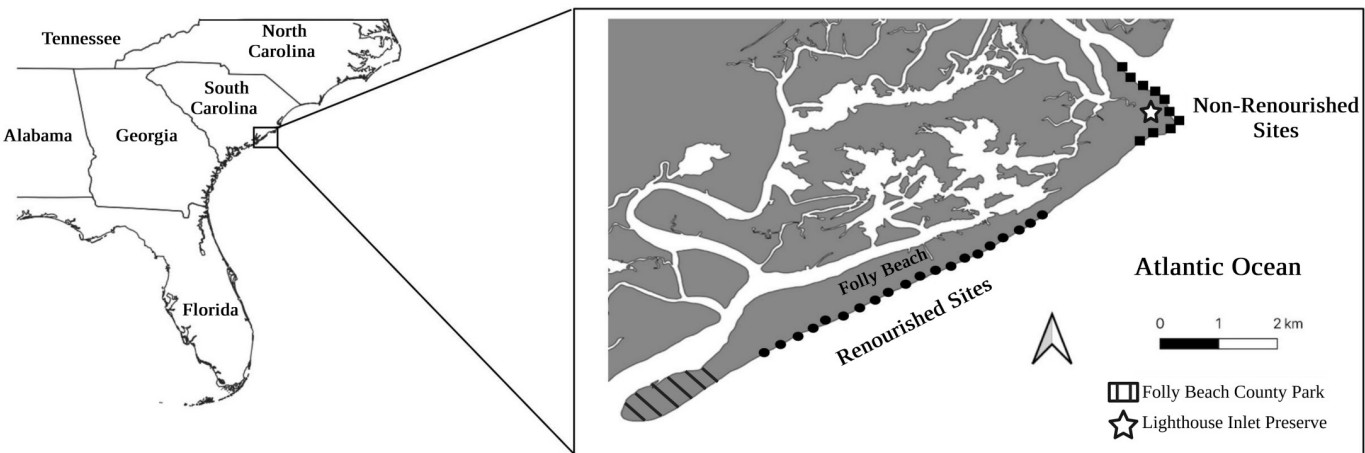

**Fig 1. Field site map.** Map of Folly Beach, SC, USA, where field sites were located. Field sites with circles indicate renourished areas, while those with squares indicate non-renourished areas. The Inset map shows South Carolina in reference to the southeastern U.S. coastline. This map was created using ArcGIS software from ESRI. [38] The basemap is supported by ESRI and licensed under the ESRI Master License Agreement. Copyright ESRI 2024. All rights reserved. Available from: https://services.arcgisonline.com/ArcGIS/rest/services/Canvas/World_Light_Gray_Base/MapServer. Edited after creation with BioRender.com.

of Georgia and South Carolina, and experiences daily tidal fluctuations >1.5 m. The island is bounded by Morris Island to the north, James Island to the west, Kiawah Island to the south, and the Atlantic Ocean along the beachfront to the eastern side. The renourished section of the beach includes all areas from the Folly Beach County Park, on the southwest side of the island, to Lighthouse Inlet Heritage Preserve at the end of Ashley Avenue on the island's northeast side. The non-renourished segment of the beach faces neighboring Morris Island and includes the area along Lighthouse Creek to Lighthouse Inlet Heritage Preserve on the northeast side of the island (Fig 1).

Folly Beach has experienced erosion issues since the installation of jetties in Charleston Harbor in 1896, which have diverted sand away from the beach [40]. Estimates of the rate of erosion at Folly Beach have ranged from 0.3 m/yr. to 1.8 m/yr [41]. Since installing the jetties, the island has undergone periodic renourishment projects aimed at protecting the beach and preserving surface area. For the renourishment project described here, 1.42 million $m^3$ of sediment was deposited along an 8.6 km portion of the beach from January to June 2014. Sediment was dredged from offshore (approximately 1.6 km of the coast) and relocated along the beach in accordance with plans outlined by the U.S. Army Corps of Engineers Folly Beach environmental assessment [42]. Notably, this project utilized sediment from four borrow areas previously used to supplement Folly Beach, all of which were exhausted following this project. Since 1994, a total of 641 core sediment samples have been collected from the established borrow areas around Folly Beach. Federal and State Agencies have used these samples to compare each area to the existing beach sediment and determine its suitability for use as renourishment fill. A complete geological analysis of these areas relative to Folly Beach is available in the U.S. Army Corps of Engineers Folly Beach Geotechnical Assessment [43].

Field sites were located on publicly accessible beaches located in Folly Beach, SC, USA. No permits are required by the city to access or collect sand and water samples except in Folly Beach County Park, which was not included in our study area. We did not sample or collect any protected species, and field and laboratory protocols followed the safety policies of Winthrop University.

## Field sampling methodology

To investigate the effects of renourishment on *E. coli*, we divided the beach into two groups: the non-renourished sections and the renourished sections. Overall, we sampled 30 sites in these two groups, with 20 in the renourished and 10 in the non-renourished (Fig 1). The 20 locations within the renourished sections were chosen based on block markers of Ashley Avenue, which runs adjacent to the beach. The ten non-renourished sections were chosen based on preexisting beach markers along the northeast portion of the beach. Samples were collected over the course of two days in June 2014 and 2015 respectively, with collection taking place during the daylight low tide period.

Each of the 30 sites in the renourished and non-renourished sections were then divided into zones based on surf exposure. Zone categories consisted of the dunes, intertidal, and subtidal zones. The dunes are the sands farthest away from the water line that would not be exposed to the surf under routine spring high tide conditions. The intertidal zones are intermediate areas of the beach that are exposed or submerged depending on the time of day. The subtidal zone includes regions of sand constantly under the surface of the water. At each zone two samples were collected, which gave a total raw data size of 240 sand samples. This procedure was employed in both 2014 (the year of renourishment) and 2015 (the year following renourishment). Sand samples were collected using 50 mL sterile conical vials, each filled with sand up to the 50 mL line. The two samples at each zone were taken randomly by skimming over the surface of the beach at a superficial depth (~0–5 cm) until 50 mL were gathered.

## Enumeration of *Escherichia coli* across field sites

The presence and abundance of *E. coli* was determined using ColiPlate™ kits manufactured by Bluewater Biosciences (Mississauga, Ontario, Canada) and were used according to the manufacturer's instructions. Briefly, the ColiPlate™ is a standard 96-well microtiter plate where each well is coated with a fluorogenic substrate, allowing for the quick enumeration of bacterial numbers [44,45]. Previous studies have shown the efficacy of defined substrate technologies (DST) and their reliability in testing for *E. coli* in water samples. DSTs are known to be comparably sensitive to traditional membrane filtration methods [46], and similar results have been observed using ColiPlates™ [47,48]. Further, defined substrate detection and enumeration methods for coliforms and *E. coli* have been widely accepted by various regulatory and scientific entities [48,49]. ColiPlate™ kits enumerate and quantify *E. coli* abundance but do not differentiate between species, as is the case with traditional membrane filtration methods used in recreational water monitoring.

To prepare samples for the ColiPlate™, 3.5 grams of sand were removed from each of the two samples from a given zone and combined in a sterile conical vial to create a composite sample. Sterile distilled water (35 mLs) was then added to the mixture. To release possible bacterial colonies, mixtures were hand-shaken for two minutes and then allowed to settle for 30 seconds before being pipetted into the ColiPlate™. This method was chosen based on prior research suggesting that 2 minutes of shaking and 30 seconds of rest were the optimal time to process samples [50]. After shaking, 300 μL aliquots were dispensed into each well and then incubated using a portable incubator at 35°C for 24 hours. Using the manufacturer's specifications, *E. coli* abundance was enumerated based on the number of wells on a plate that turned blue and fluoresced under long-wave UV light [51]. These values are reported as the most probable number of colony forming units (CFU) without confidence intervals and are based on Thomas's model [52].

### Grain size analysis

We collected 10g of sand from each of the sediment samples from each beach zone and placed it into a drying oven for 24 hours to remove moisture. After drying, the samples were sorted into different grain sizes using a standard sand sieve kit with five sets of sieves. The sieves divided the grains into the following categories: >2 mm: Very Coarse Sand; 0.5–2 mm: Coarse Sand; 0.25–0.5 mm: Medium Sand; 0.15 to 0.25 mm: Fine Sand; and 0.1 to 0.15 mm: Very Fine Sand. This followed a modified version of the protocols used in the U.S. Army Corps of Engineers manual [53].

### Statistical testing

We examined how the abundance of *E. coli* varied in 2014 and 2015 using two-way ANOVA and pairwise t-tests. Data was transformed using log + 1 in order to measure differences across sampling more accurately and prevent skewing due to very high CFU measurements in a few samples. To determine if there were relationships between the physical characteristics of the sand and the mean probable number of *E. coli* in both renourished and non-renourished sections, we used an ANOVA and linear regressions. Lastly, to compare if there were differences between sand grain sizes in renourished versus non-renourished sections of the beach, we used an ANOVA and pairwise t-tests. All statistical analyses were performed using JMP version 11 [54], with graphs made in R [55] using the ggplot2 package [56].

## Results and discussion

### Quantifying the presence and abundance of *E. coli* at renourished and non-renourished sites

In 2014 and 2015, *E. coli* was found in every zone of the beach, regardless of renourishment status (Fig 2). However, no viable *E. coli* was discovered in any of the collected water samples, regardless of renourishment status. These findings support research that suggests that *E. coli*

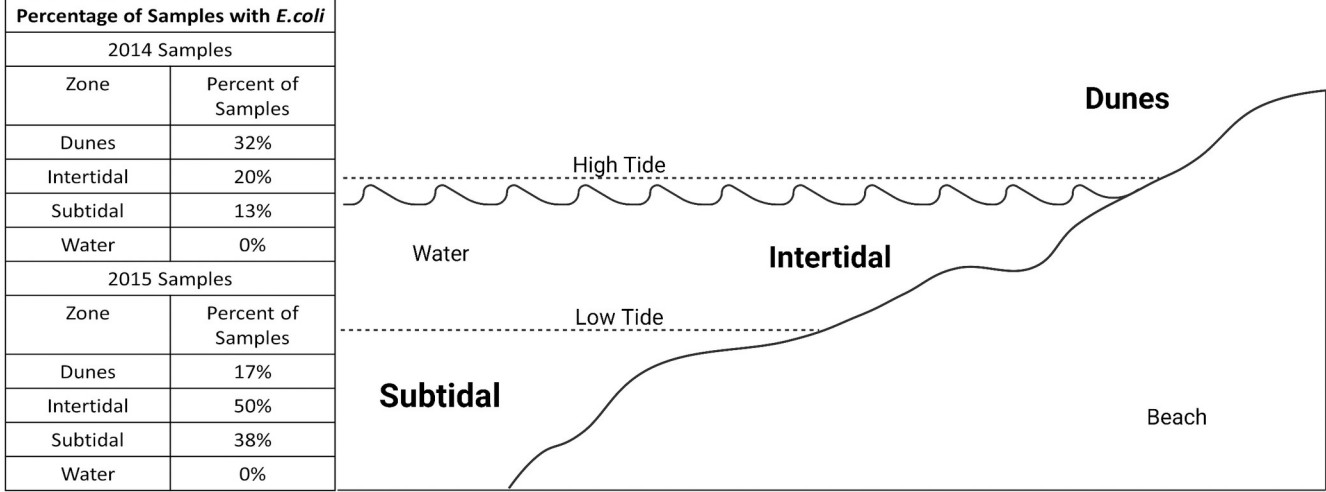

| Percentage of Samples with *E.coli* | |
|---|---|
| 2014 Samples | |
| Zone | Percent of Samples |
| Dunes | 32% |
| Intertidal | 20% |
| Subtidal | 13% |
| Water | 0% |
| 2015 Samples | |
| Zone | Percent of Samples |
| Dunes | 17% |
| Intertidal | 50% |
| Subtidal | 38% |
| Water | 0% |

**Fig 2. Beach zones and *E. coli* abundance.** Each sampling site has three zones: subtidal, intertidal, and dunes. These are determined by the height of the water surface during low and high tide. This figure shows a heuristic diagram of the zones of the beach. Sampling was conducted during low tide. Percentages represent the percentage of samples each year that tested positive for *E. coli*, regardless of whether the sample came from the beach's renourished or non-renourished area. Created with biorender.com.

can be commonly found and persist in marine sand sediments despite not being present in the water column [26,28,29,31]. Presumably, this is due to the microenvironment in the sediment providing a refuge for bacterial populations from seawater. For example, *E. coli* and entero-cocci have been shown to form sediment associated biofilms, granting them increased access to nutrients in the sediment and heightened physical protection from environmental stressors [57–60].

To evaluate the potential impact of renourishment on the presence and abundance of *E. coli* on the beach in 2014, we used a two-way ANOVA to compare the most probable number of colony forming units (CFU) of each sample across our three beach zones. CFU data was log transformed before conducting these analyses. Using this approach, we found a significant whole model effect ($F_{5,88} = 9.144$; $P = < .0001$) as well as significant differences based on beach zone ($F_{1, 83} = 17.325$; $P = < .0001$). We found that sub-tidal zones had significantly more *E. coli* when compared to the intertidal ($t = 2.919$, $P = 0.0045$) and the dunes ($t = 5.884$, $P = < .0001$; Fig 3). Additionally, we detected a significant interactive effect of both renourishment status and beach zone ($F_{1, 83} = 4.312$; $P = 0.0165$). However, no significant differences were found in 2014 when accounting for renourishment status alone ($F_{1, 83} = 2.891$; $P = 0.0928$).

To further examine the interaction of renourishment status and beach zone for the 2014 data, we conducted a series of post-hoc student's t-tests to compare mean CFU found in samples at identical zones of the beach, but which differed in their renourishment status. Here, we observed that only the intertidal zone of the beach had a significant difference between the renourished and non-renourished samples ($t = 3.099$, $P = 0.0026$), with the renourished sections having significantly higher mean CFUs of *E. coli* (Fig 3). This supports the idea that the intertidal zone, where the dredged sand is being laid, is being impacted most by

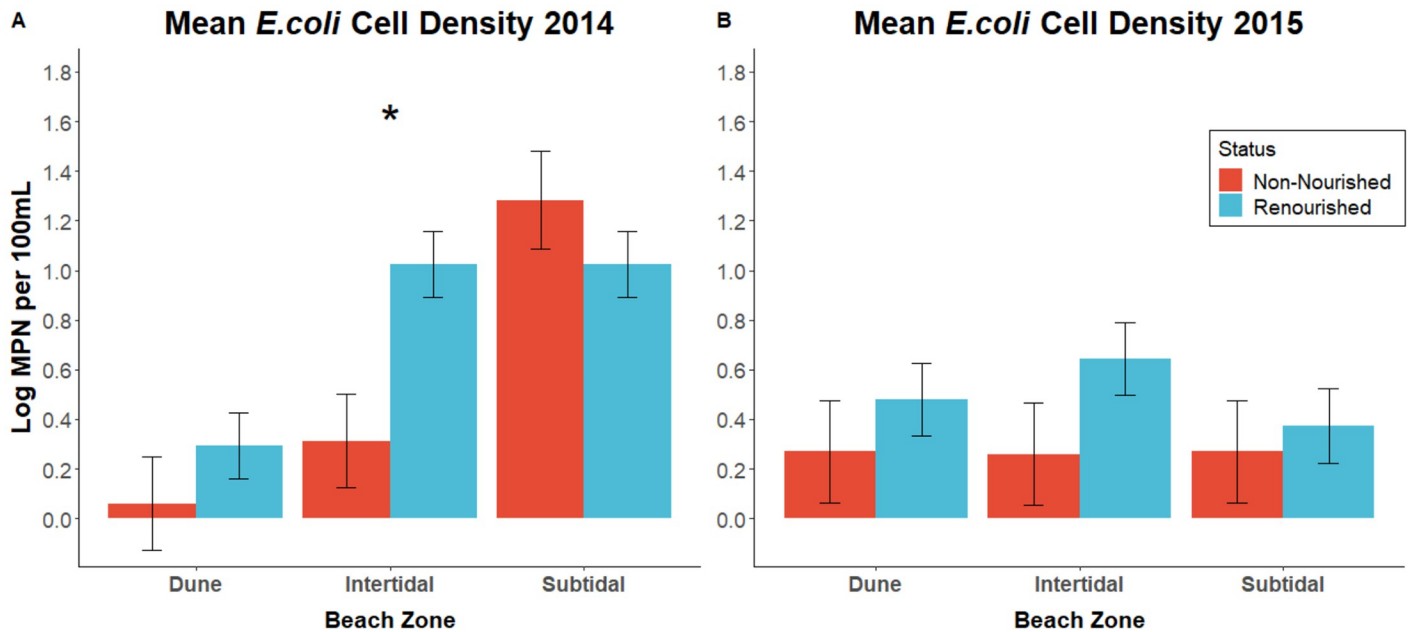

**Fig 3. *E. coli* abundance across the beach.** Comparisons of *E. coli* abundance in 2014 and 2015 across each zone of the beach categorized by renourishment status. Statistically significant results are notated with an asterisk. * = *P*<0.05. Error bars represent the standard error of the group. A. In 2014, there was significantly more *E. coli* in the subtidal zones of the beach compared to the intertidal and dunes (t = 2.919, P = 0.0045; t = 5.884, P = < .0001). However, only the intertidal zones observed differences based on renourishment status. B. In 2015, the zone differences found in 2014 had dissipated, but there was generally more *E. coli* in the renourished sections of the beach when compared to the non-renourished sections, irrespective of zone (F = 4.137; P = 0.0408).

renourishment. This could be due to various factors, ranging from physical changes to the beach to ecological changes in the local bacterial community, which facilitate more growth. However, the exact mechanisms underlying the changes caused by the renourishment disturbance remain unclear.

We then quantified how *E. coli* presence and abundance levels varied at our 30 study sites in 2015, one year after the renourishment project ended. To do this, we used a two-way ANOVA, as was done to analyze the 2014 data. We found that the differences observed in the 2014 samples were absent in the 2015 samples. We detected no whole model significance ($F_{5,88}$ = 1. 350, $P$ = 0.2517), no significant difference between the zones of the beach ($F_{1, 83}$ = 0.550; $P$ = 0.5791), and no significant interactive effect of zone and renourishment status ($F_{1, 83}$ = 0.465; $P$ = 0.6297). However, unlike in 2014, we did observe a significant difference between the renourished and non-renourished sections of the beach, with the renourished sites having comparatively more *E. coli* ($F_{1, 83}$ = 4.137; $P$ = 0.0408). We then ran post-hoc student's t-tests to compare differences between renourished sections of the beach and their non-renourished counterparts and to provide additional confirmation to the whole model test. Here, we saw no significant differences, including in the intertidal zone (Fig 3). A summary of these analyses can be found in the supplement (S1 Table).

## Quantifying differences in sand grain size at renourished and non-renourishes sites

To assess the potential impacts of renourishment on sand grain size, we used one-way ANOVAs to examine overall differences in sand grain sizes (very coarse, coarse, medium, fine, very fine) between renourished and non-renourished study sites, as well as student's t-tests to examine differences within zones based on renourishment status (S2–S4 Tables). In both 2014 and 2015, we found that there were significant differences between renourished and non-renourished sections of the beach in several zones (S3 Table). Further, the differences detected varied from 2014 to 2015.

In 2014, we found that renourished sections of the beach had a significantly higher percentage of very fine sand when compared to non-renourished sections of the beach ($F_{5, 88}$ = 39.1752, $P$ = < .0001). This is likely driven by the dune and sub-tidal zones within the renourished section of the beach containing a greater percentage of very fine sediment than their non-renourished counterparts (Fig 4). However, we also determined that the intertidal zones of renourished sections had a significantly higher percentage of very coarse and coarse grains when compared to their non-renourished counterparts. This means that while the renourished sections of the beach generally had more very fine sand grains, these dynamics were driven by large differences between the dunes and subtidal zones of renourished areas and non-renourished areas. Within the intertidal, where sand is directly deposited, there was a general shift toward coarse and very coarse grains.

In 2015, we found that renourished sections of the beach again contained significantly more very fine sand than non-renourished sections. ($F_{5, 88}$ = 25.8260, $P$ = < .0001). Further, we again detected significantly more very fine sand in the subtidal zone of renourished areas when compared to their non-renourished counterparts (Fig 4). The intertidal zones of renourished sections also contained a higher percentage of coarse grains when compared to the intertidal zones of non-renourished sections, again mimicking the results of 2014. However, there were also numerous differences between sand grain abundance in 2014 and 2015. Firstly, there were no differences recorded between grain sizes in renourished dunes and non-renourished dunes. Secondly, while the renourished intertidal zones did have more coarse sediment than non-renourished sections, they did not have more very coarse sand. Lastly, the renourished

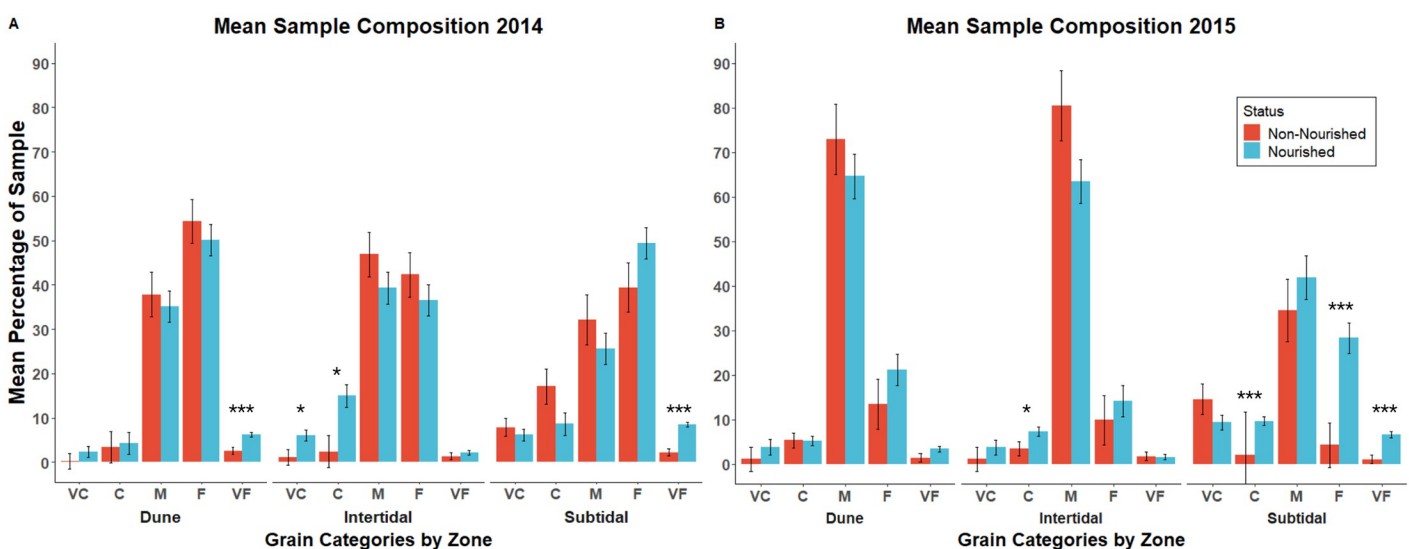

**Fig 4. Relative frequency of sand grain sizes.** Comparisons of grain sizes of sand in 2014 and 2015 across each zone of the beach categorized by renourishment status. The five grain size categories are very coarse (V.C.), coarse (C), medium (M), fine (F), and very fine (V.F.). Statistically significant results are notated with an asterisk. * = P<0.05, ** = P<0.005, *** = P<0.0005, and indicate differences at a subsection of the beach based on renourishment status. Error bars represent the standard error of the group. In 2014 and 2015, we found a statistically significant difference in the percentage of Very Fine grains found in the renourished vs. the non-renourished sections, irrespective of beach zone.

subtidal zones maintained higher percentages of very fine sand but also had significantly more fine and coarse sized grains.

These sand grain data suggest that despite established protocols recommending that transplanted sand be similar in grain size to existing sediment [61], actual grain sizes used in projects may vary depending on the individual logistics of a given project. Indeed, a previous study found that renourishment projects that took place at Folly Beach in 2005 and 2007 caused long term changes in the sediment composition of the renourishment borrow areas (where sediment is being removed from), causing a general shift toward more fine materials compared to nearby sediment not within the borrow area [62]. This study supports that finding and adds greater context, suggesting that coarse sediments may comprise a greater percentage of the overall material being moved to the beach and can remain in the intertidal zone. However, this may be an artifact of the characteristics of larger sediment, as research has shown that coarser sediment is more likely to remain on the beach over time.

### Examining how renourishment may impact *E. coli* and sand grain size

In 2014, following the renourishment event, we found significantly higher abundances of *E. coli* in the intertidal zone at renourished study sites (Fig 3). However, this effect dissipated in 2015. These findings are interesting for several reasons. First, the intertidal zone is where sand is added directly during the renourishment process. Thus, if renourishment affects *E. coli* abundance, it would most likely impact this zone of the beach. Second, these findings suggest that oceanic sand dredged from offshore (as was the case in this study where sand was extracted from approximately 1.6 km off the coast) may be a source of *E. coli* or a source of factors conducive to their growth. This is supported by observations that the renourished intertidal zone had significantly more *E. coli* immediately following the renourishment project in 2014 but not in 2015. This conclusion may not be surprising, given that in other ecosystems, *E.*

*coli* can be found in biofilms in sediment at higher levels than in the water column itself [57,58], and sediment nutrients can facilitate bacterial persistence [25,63,64].

While we detected a difference between the mean *E. coli* levels in the renourished sections of the beach and the non-renourished sections of the beach in 2015, the lack of within zone (subtidal, intertidal, and dunes) differences makes identifying the impacts of renourishment difficult. This is especially difficult given that we did not see any overall difference in *E. coli* levels between renourished and non-renourished sections of the beach in 2014. Beaches are commonly inundated with *E. coli* from various sources, ranging from anthropogenic runoff [64,65] to direct depositions by a variety of vertebrates [66,67]. If the renourishment process itself was impacting the levels of *E. coli* found, we would expect to see these differences manifest in the areas where the sand is being placed directly (the intertidal zone) and not indiscriminately across the renourished sites in every zone. Thus, it is possible our *E. coli* counts reflect differential anthropogenic or animal activity versus actual differences between the sediment microenvironments. We also would have expected these results to appear immediately following renourishment, and not to be limited to the summer following the event. As we only sampled directly after the renourishment event and the subsequent summer, we cannot rule out any other potential drivers of microbial community change. Thus, increasing the temporal scope of this type of study would likely clarify any finer scale differences between renourished and non-renourished sections more effectively. Lengthening the study scope to include different seasons, or additional years of summer sampling, could also elucidate whether Folly Beach is prone to natural fluctuations within the summer or over the course of the year, although this does not limit our ability to compare between summers in this study.

Despite observing differences in sand grain size between renourished and non-renourished sections, there was limited evidence for any relationship between grain size percentage and *E. coli* abundance (S1 Table). We evaluated the relationship between sand grain sizes (the percentage of each size category found in each sample) and the average CFU of *E. coli* in both years and found one significant correlation in each year. Each of these observations was recorded in a different portion of the beach and in different grain categories. In 2014, we found a single significant correlation between the average CFU of *E. coli* and very coarse sand in the non-renourished zone ($R^2 = 0.39$, $P < 0.005$). While in 2015, we found one significant correlation between *E. coli* and coarse sand in the renourished zone ($R^2 = 0.24$, $P = 0.03$).

The lack of a clear statistical pattern suggests that oceanic beaches are complex environments and likely have numerous factors impacting the presence and abundance of *E. coli*. For example, as previously mentioned, it may not be grain size alone influencing *E. coli* populations, but rather sand being deposited from offshore sinks where *E. coli* levels are higher or have factors conducive to bacterial growth. Research has also suggested that beach geomorphology may be a predominant influence on bacterial counts [68]. Thus, changes between areas may be due to differences in the geomorphological microenvironment across the beach. However, previous studies are inconclusive on how grain size may influence the growth of *E. coli*, and more research is needed [17,58,69]. For example, a different study evaluating the impact of renourishment on beaches found no clear link between the size of sand and the regulation of microbiological quality but did find differences between natural and artificial beaches [32]. Those findings contradicted other studies that have found that sediment morphology and particle size generally influence microbial community growth [68,70,71]. For instance, researchers studying the growth of bacteria in fertilizers of different particle sizes found that smaller particle sizes supported increased bacterial diversity irrespective of the type of fertilizer [72]. This is illustrative of both the difficulties associated with characterizing bacteria in these environments, as higher microbial loads at artificial beaches could also be due to increased tourist or animal traffic, and how microbial dynamics on beaches could change based on

microenvironment geomorphology. Lastly, beach sampling can be highly variable due to changing environmental conditions, and these phenomena can affect results [73]. This is potentially reflected in the overall differences in *E. coli* abundance observed between zones of the beach in 2014 and between renourished and non-renourished sections of the beach as a whole in 2015 (S1 Table).

Finally, our findings suggest that while renourishment may impact mean sand grain size and the presence and abundance of *E. coli*, its impacts may be short-lived. This could be due to various factors, including die-off of bacteria within the transplanted sediment, or the gradual return of bacterially-relevant nutrients in the sediment microenvironment to pre-renourishment levels. Nonetheless, these results are akin to other studies examining the implications of renourishment for microbial species. For example, a previous study by Hernandez et al. 2014 found that renourishment at a marine beach in Florida decreased surface levels of *Enterococcus spp*. over 80 days before leveling off [15]. In addition, research evaluating the effect of this renourishment project on Folly Beach, SC, USA, showed that benthic microalgal assemblages were impacted for almost six months before returning to pre-renourishment levels [16]. This gives an indication of the resilience of these environments, which is defined as the extent to which a beach system can recover from a major shock or disturbance, such as a storm event or renourishment project [74]. Abiotic and biotic components of the beach likely have different levels of resilience, and these should be considered separately when evaluating potential impacts on coastal beaches from renourishment.

## Conclusion

Since this renourishment project in 2014, our understanding of the importance of beach sediment as a microbial reservoir has increased [17,19,70,75]. This observational study adds to that understanding by supporting the idea that *E. coli* can be commonly found in sediment at marine beaches. It also adds to the literature detailing the multitude of ways renourishment can impact marine coastal ecosystems. Collectively, these findings show the impact renourishment has on beach sediment and suggest that renourishment may have consequences for the microbial ecology of marine beaches. The potential for renourishment to impact the microbial communities of beaches is particularly interesting given the recent focus on beach sediment as a microbial reservoir and a vector of human microbial infections [18,19,76]. However, more research is needed to determine how long these impacts last, the degree to which they are driven by the event of renourishment itself or the interaction of renourishment with other factors, and whether renourishment has similar impacts on other members of the beach microbial community.

## Supporting information

**S1 Table. *E. coli* ANOVA.** Summary of *E. coli* Two-Way ANOVAs with corresponding F-ratios and P values.
(PDF)

**S2 Table. Sand grain ANOVA.** Summary of ANOVA Results for Sand Grain Size Analysis with F and P values.
(PDF)

**S3 Table. Sand grain comparisons 2014.** Summary of sand grain analyses for 2014 samples. The number represents the percentage of the total sand sample belonging to each grain size category. P-values in bold symbolize statistically significant results in the t-tests between the

sections.
(PDF)

**S4 Table. Sand grain comparisons 2015.** Summary of sand grain analyses for 2015 samples. The number represents the percentage of the total sand sample belonging to each grain size category. P-values in bold symbolize statistically significant results in the t-tests between the sections.
(PDF)

## Acknowledgments

We would like to thank Dr. Cynthia Tant of Winthrop University for her insight on the project. Additionally, we would like to thank Dr. Lance Waller of Emory University for his insights on analyzing our data, Dr. Paul Turner of Yale University for feedback during editing, and the Morran lab at Emory University for their assistance in revising the manuscript.

## Author Contributions

**Conceptualization:** Victoria J. Frost, Matthew J. Heard.

**Data curation:** Jordan A. Lewis.

**Formal analysis:** Matthew J. Heard.

**Funding acquisition:** Matthew J. Heard.

**Investigation:** Jordan A. Lewis, Victoria J. Frost, Matthew J. Heard.

**Methodology:** Victoria J. Frost, Matthew J. Heard.

**Project administration:** Victoria J. Frost.

**Resources:** Matthew J. Heard.

**Supervision:** Victoria J. Frost, Matthew J. Heard.

**Validation:** Jordan A. Lewis, Matthew J. Heard.

**Visualization:** Jordan A. Lewis, Matthew J. Heard.

**Writing – original draft:** Jordan A. Lewis, Victoria J. Frost, Matthew J. Heard.

**Writing – review & editing:** Jordan A. Lewis, Victoria J. Frost, Matthew J. Heard.

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
