## [Decision Letter · Decision Letter 0]

26 Feb 2024

PONE-D-24-00797Examining the potential impacts of a coastal renourishment project on the presence and abundance of * Escherichia coli *PLOS ONE

Dear Dr. Lewis,

Thank you for submitting your manuscript to PLOS ONE. After careful consideration, we feel that it has merit but does not fully meet PLOS ONE’s publication criteria as it currently stands. Therefore, we invite you to submit a revised version of the manuscript that addresses the points raised during the review process.

**Please submit the revised version.**==============================

We look forward to receiving your revised manuscript.

Kind regards,

Md. Asaduzzaman Shishir, PhD

Academic Editor

PLOS ONE

Additional Editor Comments:

It is recommended that the authors amend the manuscript following the comments made by the reviewers.

Reviewers' comments:

Reviewer's Responses to Questions

**Comments to the Author**

1. Is the manuscript technically sound, and do the data support the conclusions?

Reviewer #1: Yes

Reviewer #2: Partly

2. Has the statistical analysis been performed appropriately and rigorously? 

Reviewer #1: Yes

Reviewer #2: Yes

3. Have the authors made all data underlying the findings in their manuscript fully available?

Reviewer #1: Yes

Reviewer #2: Yes

4. Is the manuscript presented in an intelligible fashion and written in standard English?

Reviewer #1: Yes

Reviewer #2: Yes

5. Review Comments to the Author

Reviewer #1: Please mention the exact time/month of the sampling on page 7 line 144.

Is there any seasonal influence on the presence and abundance of E. coli around beach area? How may the authors relate seasonal influence to this study?

Reviewer #2: 1. As the impact of renourishment project was examined in terms of E. coli and its presence or absence, it might be insightful if the type of E. coli whether pathogenic or non-pathogenic was also detected.

2. The author suggests that following renourishment project the changes between microbial community and physical changes of the beach might be for short time. What might be the underlying reasons? Also required experimental evidence.

3. It would be intuitive if the work is conducted for longer period time (more years). Also is there any seasonal impact on microbial community there?

4. The asterisk should be in place in the graph.

6. PLOS authors have the option to publish the peer review history of their article (what does this mean?). If published, this will include your full peer review and any attached files.

Reviewer #1: **Yes: **Umme Tamanna Ferdous

Reviewer #2: No

---

## [Author Response · Author response to Decision Letter 0]

7 Mar 2024

Manuscript Number: PONE-D-24-00797

Title: Examining the potential impacts of a coastal renourishment project on the presence and abundance of Escherichia coli

1. Please ensure that your manuscript meets PLOS ONE's style requirements, including those for file naming. The PLOS ONE style templates can be found at……………

Authors Response: We have gone through the manuscript and insured that the paper meets the style requirements of PLOS ONE. 

Authors Response: In the submission portal, we do not see distinct sections named Funding Information’ and ‘Financial Disclosure. Thus, we have added a section to the manuscript file called “Funding Sources” in order to compliment the financial disclosure section in the portal.

Authors Response: We have made the data associated with this manuscript available online via Dryad. The data can be found at https://datadryad.org/stash/dataset/doi:10.5061/dryad.zgmsbcchp

4. We note that Figure 1 in your submission contain [map/satellite] images which may be copyrighted. All PLOS content is published under the Creative Commons Attribution License (CC BY 4.0), which means that the manuscript, images, and Supporting Information files will be freely available online, and any third party is permitted to access, download, copy, distribute, and use these materials in any way, even commercially, with proper attribution. For these reasons, we cannot publish previously copyrighted maps or satellite images created using proprietary data, such as Google software (Google Maps, Street View, and Earth).

Authors Response: We have added the copyright information for the map used in figure 1 on page 5, lines 88-95. The map is available for use through ArcGIS.

Reviewer's Responses to Questions

Comments to the Author

1. Is the manuscript technically sound, and do the data support the conclusions?

Reviewer #1: Yes

Reviewer #2: Partly

2. Has the statistical analysis been performed appropriately and rigorously?

Reviewer #1: Yes

Reviewer #2: Yes

3. Have the authors made all data underlying the findings in their manuscript fully available?

The PLOS Data policy requires authors to make all data underlying the findings described in their manuscript fully available without restriction, with rare exception (please refer to the Data Availability Statement in the manuscript PDF file). The data should be provided as part of the manuscript or its supporting information or deposited to a public repository. For example, in addition to summary statistics, the data points behind means, medians and variance measures should be available. If there are restrictions on publicly sharing data—e.g. participant privacy or use of data from a third party—those must be specified.

Reviewer #1: Yes

Reviewer #2: Yes

4. Is the manuscript presented in an intelligible fashion and written in standard English?

Reviewer #1: Yes

Reviewer #2: Yes

Authors Response to prompts I-IV: The authors would like to thank the reviewers for their time and effort reviewing the manuscript for publication in PLOS ONE. Please see specific responses to reviewer feedback below.

5. Review Comments to the Author

Reviewer #1: 1. Please mention the exact time/month of the sampling on page 7 line 144.

2. Is there any seasonal influence on the presence and abundance of E. coli around beach area? How may the authors relate seasonal influence to this study?

Authors Response to Reviewer I:

1. We have added the exact time and month of sampling in both years of our study to page 7, lines 139-141.

2. Seasonal influences may impact the presence and abundance of E. coli at this beach, as previous work has shown that levels of fecal indicators can fluctuate during the year. However, in this study we evaluated the beach during the same season and month of the year in back-to-back years, so this does not limit our ability to make comparisons between the summers. However, these fluctuations are a consideration and we have added brief conversation of this on page17, lines 351 to 354.

Reviewer #2: 

1. As the impact of renourishment project was examined in terms of E. coli and its presence or absence, it might be insightful if the type of E. coli whether pathogenic or non-pathogenic was also detected.

2. The author suggests that following renourishment project the changes between microbial community and physical changes of the beach might be for short time. What might be the underlying reasons? Also required experimental evidence.

3. It would be intuitive if the work is conducted for longer period time (more years). Also is there any seasonal impact on microbial community there?

4. The asterisk should be in place in the graph.

Authors Response to Reviewer II:

1. We agree that determining the serotypes of E. coli isolated on the beach, and whether they are pathogenic or not, would add more clarity to this study. However, we chose the methods we did to attempt to approximate standard FIB monitoring procedures on a bacterium usually not tested for, in an environment where standards would typically be waived. FIB monitoring is inherently limited by its focus on abundance and distribution, and not serotype or pathogenicity. Our ultimate goal was to investigate the degree to which we could find E. coli in the sand, and if those numbers changed based on renourishment. As normal monitoring for enterococci (as is typically measured in marine environments) would not have indicated whether these strains were pathogenic or not before leading to a beach closure, we sought to use similar methods and compare our results. However, this limitation is still important and we now clarify that the ColiPlate™ does not identify strain or pathogenicity on page 8 lines 165-167. 

2. We sampled the same locations on one beach in back-to-back years, with not relevant changes to the beach outside of the renourishment project. Ultimately, we are limited in our ability to directly address mechanisms behind this change without experimentation. However, we analyzed sand grains in an attempt to identify a mechanism given existing research on the effects of the sediment micro-environment on bacteria. Research has shown that biofilm formation and access to nutrients can be important for bacterial survival in these environments, as well as the normal competition, predation, and parasitism experienced by bacteria that could change with environmental perturbations (like a renourishment project). We discuss these possibilities throughout and attempt to find answers in the literature, however we now also provide more direct discussion on page 18 lines 390-392.

3. Please see the comment to Reviewer I above. In terms of the length of study, we agree more conclusions could be made if this had been done for longer than 2 years. However, this study was covered by a grant which lasted only two years and was conducted by someone who, at the time, was an undergraduate student and was limited in their ability to reassess the beach. These are limitations of the study and we have added note of these shortcomings on page 17, lines 341- 354. We also acknowledge that to fully understand the microbial community and how it is impacted, research must move past model species. We have included this on line 415 of page 19.

4. We have rechecked all of our figures and images and see all of the asterisk in place as we intended. We are unsure what the reviewer meant by comment 4, or if this reflects an error in the image they downloaded during review.

---

## [Decision Letter · Decision Letter 1]

7 May 2024

Examining the potential impacts of a coastal renourishment project on the presence and abundance of * Escherichia coli *

PONE-D-24-00797R1

Dear Dr. Lewis,

We’re pleased to inform you that your manuscript has been judged scientifically suitable for publication and will be formally accepted for publication once it meets all outstanding technical requirements.

Kind regards,

Md. Asaduzzaman Shishir, PhD

Academic Editor

PLOS ONE

Additional Editor Comments (optional):

Reviewers' comments:

Reviewer's Responses to Questions

**Comments to the Author**

1. If the authors have adequately addressed your comments raised in a previous round of review and you feel that this manuscript is now acceptable for publication, you may indicate that here to bypass the “Comments to the Author” section, enter your conflict of interest statement in the “Confidential to Editor” section, and submit your "Accept" recommendation.

Reviewer #1: (No Response)

Reviewer #2: All comments have been addressed

2. Is the manuscript technically sound, and do the data support the conclusions?

Reviewer #1: (No Response)

Reviewer #2: Yes

3. Has the statistical analysis been performed appropriately and rigorously? 

Reviewer #1: (No Response)

Reviewer #2: Yes

4. Have the authors made all data underlying the findings in their manuscript fully available?

Reviewer #1: (No Response)

Reviewer #2: Yes

5. Is the manuscript presented in an intelligible fashion and written in standard English?

Reviewer #1: (No Response)

Reviewer #2: Yes

6. Review Comments to the Author

Reviewer #1: (No Response)

Reviewer #2: (No Response)

7. PLOS authors have the option to publish the peer review history of their article (what does this mean?). If published, this will include your full peer review and any attached files.

Reviewer #1: **Yes: **Umme Tamanna Ferdous

Reviewer #2: **Yes: **Dr. Tamanna Zerin

---

## [Editor Report · Acceptance letter]

16 May 2024

PONE-D-24-00797R1 

PLOS ONE

Dear Dr. Lewis, 

I'm pleased to inform you that your manuscript has been deemed suitable for publication in PLOS ONE. Congratulations! Your manuscript is now being handed over to our production team.

Kind regards, 

on behalf of

Dr. Md. Asaduzzaman Shishir 

Academic Editor

PLOS ONE